# OpenReview forum: "Compromised Turing Machines: Adversarial Interference and Endogenous Verification"
_ICLR.cc/2025/Conference — Submitted to ICLR 2025_

### Official Review · Reviewer_q56N · 2024-10-26

**Soundness:** 3
**Presentation:** 3
**Contribution:** 2
**Rating:** 3
**Confidence:** 3

**Summary:**

This paper presents the concept of a compromised Turing Machine (CTM), in which an adversary can manipulate both the tape and the internal state in between discrete time steps. The authors show that no internal verifier to the CTM can correctly verify that the state and/or tape have not been tampered with. The paper draws parallels to Descartes' deus deceptor, and argue that the only way to get around this impossibility result is through an external source of truth (as was the case for the deus deceptor). The authors propose a set of third party verification techniques that can be used for verification.

**Strengths:**

This paper was fun to read. I appreciated learning about the deus deceptor, and seeing the parallels drawn to the presented CTM model. I also think the CTM is a natural computational model to consider, so it was nice that the authors formalized this idea.

**Weaknesses:**

My main concern is that I don't think this is a machine learning paper---I'm not sure what exactly would be the right venue, but as it stands, I don't think this is the right community or format for this work. Within the engineering/computer science discipline, the CTM seems like a construct potentially of interest to the security community. The connections to machine learning seemed quite loose to me.

Another major concern is that the article read more like a position paper than a research paper. The paper is not empirical, but it also didn't seem like a theory paper. The main theorem on the impossibility of complete endogenous verification seems to be the only technical result, and it is quite light. I don't say this to disparage the work, I think it's just that the main value of the paper was in discussing interesting new ways of modeling adversarial actions, and relating those models to the philosophy literature. I almost wonder if this article would be a better fit for a venue that accepts position papers, or white papers. To me, the technical contribution doesn't currently fit the requirements of a top research conference (at least in the field of CS/engineering).

This is minor, but there were some issues of redundancy in the writing, with parts of definitions repeated almost verbatim (e.g., the definition of a Turing machine, the explanation of a CTM)

**Questions:**

Why did you submit this work to a machine learning venue in particular?

---

### Official Review · Reviewer_3LxB · 2024-10-26

**Soundness:** 3
**Presentation:** 3
**Contribution:** 1
**Rating:** 3
**Confidence:** 4

**Summary:**

The authors provide a formal argument for why a Turing-machine cannot detect if it is compromised, where there in an actor who is able to arbitrarily edit the tape in between each step. They then outline how this might be addressed using techniques from secure computation.

**Strengths:**

The paper is fun!

It's a cute idea to talk about compromised Turing machines in a formal manner. The result also seems almost self-evidently correct.

I also appreciated the connections to Descartes, and thought it made for enjoyable reading.

**Weaknesses:**

I'm not convinced that the main and only result is insightful enough! It seems tautological to say that if a machine uses a stored value to determine if it has been tampered with or not, and an adversary can tamper with that value, then the machine can not actually detect tampering. That is fundamentally the argument being made.

There is some value in formalizing it, but it is slight in relation to a full conference paper (not that it isn't worth doing at all), and that the paper really needs additional insights or results to stand on its own.

The paper also repeats itself a little without adding major new substance (eg; when discussing Descartes).

One direction the paper could have gone to try explore the topic a little more would be to talk about stored-program computers vs Turing machines, and what it means that the adversary couldn't modify the transition function. Then tying that back to real systems and discussing what the model elucidates.

**Questions:**

What is the gain that we get in the model by assuming that even can tamper with the tape but not any other aspect of the machine?

---

### Official Review · Reviewer_DPG3 · 2024-10-30

**Soundness:** 3
**Presentation:** 3
**Contribution:** 1
**Rating:** 3
**Confidence:** 3

**Summary:**

This paper considers a model of computation in which an adversary may arbitrarily alter the computer’s internal state. They formally define such a computer as a _Compromised Turing Machine (CTM)_. A CTM is a standard Turing machine, with an additional predicate P_safe designed to (attempt to) verify the integrity of the current state and tape. An adversary Eve may alter the CTM’s tape and state after each time step. This paper proves that an adversary can always alter the CTM in a way that is undetectable by P_safe. It then compares this adversarial model to Descartes’ deus deceptor thought experiment, in which a powerful deceiver may alter one’s perception of reality. Both this paper and Descartes conclude that in the presence of such an adversary/deceptor, an external source of trust is necessary.
This paper then surveys several approaches to building trust in partially untrusted systems, using tools from cryptography, distributed systems, and verification.

**Strengths:**

This paper is clearly written, and the parallel between the CTM model and Descartes’ deus deceptor thought experiment is interesting.
Section 5 presents a variety of techniques drawing from different fields.

This paper has most potential as a survey of the ways in which one can ensure integrity of computation executed on an untrusted computer, with an interesting analogy to how one can verify what reality is more philosophically.

**Weaknesses:**

Cryptography has long considered essentially this model in the form of malicious parties that may deviate from the prescribed computation arbitrarily. The fact that a single malicious party may prevent correct computation is well known, and hence a large body of cryptographic work aims to achieve verifiable computation in relaxed models. Relevant topics from cryptography that are missing from this paper include _program checking, secure multiparty computation, delegation, authenticated data structures_, and _memory checking_. The cryptography tools that are mentioned are not clearly helpful, for example homomorphic encryption. Even if the contents of the CTM’s tape are homomorphically encrypted, Eve can always rewrite the tape and alter the state to some accepting (or rejecting, if she wishes) configuration. Doing so does not require any knowledge of the current tape contents.
Memory/program checking, which are well studied in cryptography, encompasses the goal that many of the mentioned verification techniques aim to achieve. Memory checking considers a large untrusted database, which is augmented with a very small amount of trusted storage in order to ensure integrity.
2.2 (Adversarial Models in Cryptography) should be expanded to address models in cryptography that consider internal computational integrity, for example program checking and memory checking.
5.4 (Self-correcting systems) should clarify how it relates to the impossibility in Theorem 1.

Overall, the main contribution of this paper is the introduction of the concept of a CTM, along with techniques for secure computation in such a setting. However, the concept of a CTM is not new and has been studied under the umbrella of malicious parties in cryptography. Furthermore, the discussion of techniques should be expanded to include more cryptographic techniques. While clearly written, the paper does not put forth sufficiently novel ideas. This paper could be improved by adding discussion of related adversarial models in cryptography, and updating the discussion of techniques for integrity to include those from cryptography listed above.

**Questions:**

How does your model relate to the adversarial model considered in memory checking (eg, https://link.springer.com/chapter/10.1007/978-3-642-00457-5_30)?

---

### Official Review · Reviewer_CmGd · 2024-11-04

**Soundness:** 3
**Presentation:** 3
**Contribution:** 3
**Rating:** 5
**Confidence:** 3

**Summary:**

This paper introduces the concept of Compromised Turing Machines (CTMs), where an adversary has the capability to modify both the tape and the internal state between computational steps. The authors argue that CTMs reveal significant vulnerabilities in self-verification and computational integrity in adversarial contexts. To address these vulnerabilities, they propose several approaches, including external verification and cryptographic techniques, to improve computational resilience against adversarial interference.

**Strengths:**

- The concept of CTMs is interesting.
- The paper is generally well-written.
- The philosophical analogy to Descartes' skepticism helps to make the concept more accessible.

**Weaknesses:**

- The threat model lacks clarity.
- The paper does not provide formal proofs or concrete applications.
- Lack of novelty and contributions

I n more detailed, I struggled a little with the following aspects that led to my score, essentially boiling down to the synergy between tasks, potential countermeasures against attackers, and the assumptions underlying the attack model, as outlined below:

The proposed method does not seem to account for the synergy between tasks. Specifically, an adversary might bypass defenses by performing harmful actions through a sequence of well-designed tasks. For instance, if transferring money from A to B is harmful, the model could break this action into two steps: first transferring money from A to C, and then from C to B. Overlooking task synergy could fail to detect such harmful intentions.

The core of the proposed defense is dynamically adjusting the threshold that determines whether a task is untrusted based on the model’s credibility, which is calculated from previous performance. In this case, an adversary could adapt their behavior, first establishing credibility by performing safe tasks, then gradually increasing the harmfulness of their actions while keeping them below the threshold. Eventually, the threshold might become high enough that the system perceives all actions from the adversary as harmless, despite potentially harmful intentions.

The authors assume that, alongside the untrusted strong model, the defender has access to a series of weak trusted models to assess harmfulness. However, this assumption requires further discussion. If the trusted models are too weak, the untrusted model could deceive them. Conversely, if they are too strong, it raises the question of why these trusted models are not simply used to perform the tasks directly. The experimental setting raises this concern further, as the authors use Claude-3-Opus as a "weak" trusted model, which performs almost as well as the untrusted “strong” model.

**Questions:**

The topics that would like to see clarified are in the weekness-section.

---

### Meta-Review · Area_Chair_eVsH · 2024-12-15

**Metareview:**

Thanks for submitting your work to ICLR.

The reviewers generally found the topic of this paper interesting, which studies a special type of Turing Machine with compromised states or tape segments.

Reviewers raised a number of major concerns regarding the relevance of this work to the machine learning community, a lack of comparison with the broader cryptographic literature (e.g., maliciously secure multiparty computation, or memory checking), an unclear threat model, and a rather light technical contribution.

I thus recommend rejection.

**Additional Comments On Reviewer Discussion:**

there was no rebuttal

---

### Decision · Program_Chairs · 2025-01-22

Reject